# Hepatitis C Virus Infection in Europe

**DOI:** 10.3390/pathogens13100841

**Published:** 2024-09-28

**Authors:** Margarida Simão, Cristina Gonçalves

**Affiliations:** Pediatric Gastrenterology and Hepatology Unit, Pediatric Hospital Dona Estefânia, ULS S. José, 1169-045 Lisbon, Portugal

**Keywords:** hepatitis C, direct-acting antiviral, Europe

## Abstract

The Hepatitis C Virus (HCV) is a significant public health challenge in European countries. Historically, healthcare-related procedures were the primary source of HCV infection in Europe. However, with the implementation of blood safety programs, injection drug use has become the main transmission route. The infection’s distribution and genotype prevalence vary widely across the continent. Even with the availability of highly effective direct-acting antiviral (DAA) therapies, HCV infection is far from being controlled. A significant proportion of patients remain undiagnosed, contributing to the ongoing transmission of the virus. Additionally, several barriers hinder the widespread use of DAAs, including high treatment costs, stigma, poor linkage to care, and considerable geographical variations in prevalence and transmission routes. The World Health Organization has set ambitious targets to reduce liver-related deaths, decrease new viral hepatitis infections, and ensure that 90% of infected individuals are diagnosed by 2030. However, most European countries face challenges, highlighting the need for screening programs, funding mechanisms, and public health strategies to effectively control HCV infection in Europe.

## 1. Introduction

Hepatitis C Virus (HCV) infection is a global health challenge and a leading cause of liver-related morbidity and mortality. Approximately 71 million people worldwide are chronically infected with HCV [1], with 15 million of them living in Europe [2,3]. Despite the availability of highly effective direct-acting antiviral (DAA) therapies, many people remain undiagnosed. In fact, it is estimated that only about one-third of chronically infected patients in Europe are aware of their diagnosis, highlighting a substantial hidden burden [4,5,6].

In this regard, the World Health Organization (WHO) adopted a resolution to improve the prevention, diagnosis, and treatment of viral hepatitis. The WHO’s targets include a 65% decrease in liver-related deaths, a 90% reduction in new viral hepatitis infections, and ensuring that 90% of individuals with viral hepatitis are diagnosed by 2030 [7]. Achieving these targets is crucial for the elimination of HCV, yet a comprehensive effort is necessary in Europe, as the region lags behind the United States in HCV treatment and diagnosis [4,8].

This review aims to give insights into the status of HCV infection across the continent. It will explore the epidemiological trends as well as the effectiveness of the current prevention, diagnosis, and treatment methods regarding HCV.

## 2. Methodology

This paper is a narrative review focused on HCV infection in Europe. A systematic search was conducted in databases including PubMed, Scopus, Web of Science, and Google Scholar, targeting peer-reviewed articles, clinical guidelines, reports from health organizations, and conference proceedings. The search terms used included combinations related to HCV prevalence, incidence, treatment, and epidemiology in Europe. Articles were selected based on their relevance to HCV in Europe, with a priority given to high-quality studies, systematic reviews, and meta-analyses, when available. Studies focusing on regions outside Europe were excluded. A critical review of the selected articles was conducted to ensure the accuracy and comprehensiveness of the data presented.

## 3. Epidemiology of HCV in Europe

HCV infection is a major public health issue in Europe. The WHO estimates that in the European region, between 8 and 12 million people are infected with HCV [8,9] and according to the European Center for Disease Control (ECDC), there is a prevalence of 0.5% of the population infected with HCV in the EU/EEA [10]. Currently, the infection is largely associated with people who inject drugs (PWIDs), particularly in developed countries [11], with significant prevalence also found among prisoners and HIV-positive men who have sex with men (MSM) [12]. Migration is also an important factor to consider as about 14% of the adults affected by HCV infection are migrants and 79% of migrants in Europe originate from endemic countries [13,14].

Concerning incidence, in 2022, 29 EU/EEA countries reported 23,273 cases of HCV infection. When excluding countries that reported only acute cases, the total number of cases was 23,249, resulting in a crude rate of 6.2 cases per 100,000 population [15]. The overall incidence rate rose from 6.7 per 100,000 in 2013 to 7.6 per 100,000 in 2014, then gradually declined to 6.6 in 2019. In 2020 and 2021, there was a significant drop, with the rate falling to between 4.6 and 4.7 per 100,000, followed by a 38% increase to 6.5 per 100,000 in 2022 [10]. The decrease in incidence in 2020 and 2021 reflects the impact of the COVID-19 pandemic on case detection and reporting and the limitation of migration [6,8].

European HCV cases show a ratio of 1.6 to 1.9 men for every woman. The age group most affected among males was 35–44 years, while for females it was 55–64 years [10].

Concerning HCV genotypes in Europe, genotype 1 is the most common, accounting for 64.4% of infections, followed by genotype 3 (25.5%), genotype 2 (5.5%), and genotype 4 (3.7%). Only small percentages of genotype 5, genotype 6, and mixed or not further classified genotypes are reported. Genotypes 1a and 3 are more common among PWIDs, while genotype 1b is associated with transmission through blood products [16]. In HCV–HIV co-infected populations, genotype 1 remains the most reported genotype [2].

## 4. Geographical Variations of HCV Infection in Europe

The epidemiology of HCV infection in Europe shows significant geographical variations in the prevalence rates, genotype distributions, and primary modes of transmission. These variations are influenced by factors such as healthcare practices, injection drug use, and migration patterns. On the other hand, there is considerable variation in the quality and comprehensiveness of epidemiological data. Some countries lack national HCV surveys, making accurate prevalence estimates challenging [17]. In 2016, less than half of the European (44%, n = 12) countries surveyed by the European Liver Patients Association maintained a national registry for HCV prevalence, and only 44% (n = 12) had a national registry for hepatocellular carcinoma (HCC) [18].

The Polaris Observatory HCV performed a modelling study stating that the highest prevalence of infection was in Eastern Europe (6.7 million), followed by Western Europe (2.3 million), and Central Europe (1.2 million) [19]. However, in every region and country, the prevalence is variable. In Central Europe, for instance, the prevalence is variable from 0.2% in Slovakia to up to 1.7% in Latvia [16,20], and in Southern Europe, there are areas of Italy and Greece with much higher prevalences than the rest of the region or country [16,21].

The incidence of new HCV infections has generally declined in Western and Central Europe due to effective harm reduction and healthcare practices. However, Eastern Europe continues to see high incidence rates, primarily due to the prevalence of PWIDs. The highest reported prevalence among PWIDs is in Sweden (82%) and Spain (72%), contrasting with UK (36.50%) and Croatia (34.04%) [12].

Genotype 1b is especially common in countries with historically high iatrogenic transmission rates, such as Italy and Romania. Genotype 3 is the second most common in many European countries, especially in Northern Europe, while genotype 2 holds that position in Italy [2].

Mortality rates from HCV-related liver diseases show an east–west gradient. Additionally, Western Europe has higher mortality from HCV-related HCC, while Eastern Europe has higher mortality from HCV-related cirrhosis [22].

## 5. Transmission of HCV Infection in Europe

HCV is an epidemiologically complex disease that affects several key populations, spreading through multiple transmission routes, often remaining asymptomatic, and manifesting in diverse clinical phases during chronic infection. The relative impact of the different contributors to the HCV has changed over the recent decades. In Western Europe, most HCV infections were acquired in the 1970s and 1980s, prior to the implementation of safer medical practices and HIV prevention programs. Consequently, older generations in countries like Belgium, France, and Germany have higher rates of HCV [22]. In contrast, some countries such as Greece, Turkey, and Romania have seen more recent increases in HCV infections due to the rise in injection drug use and historically unsafe medical practices [22,23].

Blood safety has improved dramatically with the introduction of blood-product screening assays in 1989 and hemovigilance systems in all EU/EEA countries leading to a significant change in the risk factors for HCV. Currently, the primary route of HCV transmission is through injecting drug use, especially among young adults, accounting for 53% of acute and 64% of chronic HCV cases. Other significant routes include nosocomial transmission and sexual transmission, particularly among MSM [10].

The implementation of HIV prevention programs in PWIDs, including increased access to sterile injection equipment, self-injection facilities, professional counseling, and substitution programs, has also successfully reduced the transmission rates of HCV among this population in Western Europe over the past decades [24].

As stated previously, migration also significantly impacts HCV epidemiology in Europe. A substantial portion of HCV-infected individuals in Western Europe, such as in Germany and The Netherlands, are migrants from countries with high HCV prevalence like Poland, Russia, and Turkey [22]. Screening strategies for these populations could help reduce the disease burden.

## 6. Diagnostic Approach—Screening and Testing in Europe

The screening and diagnosis of HCV in Europe faces significant challenges [12,25,26]. Several barriers impede effective HCV screening, including the low levels of awareness and knowledge about HCV among the general public and healthcare providers; the asymptomatic nature of chronic HCV, which discourages individuals from seeking testing; the stigma and discrimination associated with key populations, such as PWID and MSM, which can prevent them from accessing testing services; and structural barriers, such as the requirement for healthcare professionals to supervise testing, which can limit the availability of these services [27].

The European Association for the Study of the Liver (EASL) recommends an anti-HCV antibody serology test as the primary diagnostic tool for screening, indicating past or present HCV infection. If this test returns positive, a sensitive RNA test is necessary to confirm the infection. Novel approaches like dried blood spot and saliva sampling and the potential use of self-testing kits can further enhance screening by making testing more accessible [27,28].

To address the diverse transmission routes and affected populations, targeted screening is crucial, including in key populations: PWIDs, prison populations, migrants from high-prevalence countries, MSM, people with HIV, and healthcare workers. However, the WHO’s targets for 2030 emphasize the need for national screening programs that extend beyond high-risk populations [29]. Screening in various settings, such as primary care, hospitals, and community-based locations, is also very important. Community-based testing services, including those in drug services, migrant clinics, and pharmacies, have shown high testing uptake and positivity rates [27].

## 7. Treatment of HCV in Europe

### 7.1. Evolution of HCV Therapy

The evolution of HCV treatments in Europe has been marked by significant advancements and challenges. Before the advent of DAAs, HCV treatment primarily used interferon-based therapies, that had several limitations, including prolonged treatment durations, significant side effects, and relatively low cure rates. Patients frequently experienced flu-like symptoms, depression, and hematological abnormalities, which led to poor adherence and the discontinuation of therapy in many cases [30].

The introduction of the first DAAs, telaprevir and boceprevir, in 2011 and the EMEA approval of the first interferon-free therapy (Sofosbuvir) in 2014 marked a significant breakthrough in HCV treatment [31,32,33]. The development and approval of additional DAAs provided highly effective treatment options with sustained virologic response (SVR) rates exceeding 95% [34].

In 2018, the EASL recommended that all patients without contraindications should be considered for treatment with DAAs [35]. However, the high list prices of these treatments led to significant restrictions on their initial use, and governments and payers had to manage the financial impact. Consequently, many countries implemented a stepwise approach to DAA reimbursement, prioritizing treatment for patients with advanced liver disease. Despite the high costs, financing arrangements and strong leadership have gradually facilitated the removal of DAA restrictions, allowing access to more patients [36]. Currently, the state of HCV treatment in Europe is promising. With the availability of more pan-genotypic DAAs, treatment has become simpler, shorter, and more effective. Many European countries have made significant progress in eliminating restrictions on DAA treatments, enabling broader patient access [34].

### 7.2. Current Guidelines on Treatment for HCV in Europe

The EASL guidelines for HCV treatment reflect the latest advancements in therapeutic approaches. They emphasize that all patients with HCV should be considered for treatment, irrespective of disease stage, to prevent liver disease progression and transmission. The current guidelines recommend pan-genotypic DAAs as the first-line treatment for HCV due to their high efficacy, safety profile, and ease of use. These DAAs include Sofosbuvir/Velpatasvir, an effective combination across all HCV genotypes, typically administered for 12 weeks, and Glecaprevir/Pibrentasvir, another pan-genotypic regimen, prescribed for 8 weeks for treatment-naïve patients without cirrhosis [34].

For individuals with compensated cirrhosis (Child–Pugh A), treatment with the same regimens is recommended as those without cirrhosis, though sometimes with an extended duration or the addition of ribavirin. For decompensated cirrhosis (Child–Pugh B or C), Sofosbuvir/Velpatasvir plus ribavirin for 12 weeks or without ribavirin for 24 weeks is recommended. However, for patients with renal impairment, Glecaprevir/Pibrentasvir is preferred due to its safety profile, including in those on dialysis [34].

In patients with HIV co-infection, the same DAA regimens used for mono-infected patients are recommended, ensuring the careful management of drug–drug interactions.

Concerning retreatment strategies for patients who fail initial DAA therapy, they include the combination of Sofosbuvir/Velpatasvir/Voxilaprevir (for those who have failed a regimen containing an NS5A inhibitor) and Glecaprevir/Pibrentasvir or Sofosbuvir/Velpatasvir with ribavirin (based on the resistance profile and prior DAA exposure) [34].

The EASL highlights the need for shorter treatment regimens, enhanced screening and linkage to care and the use of resistance testing in certain scenarios, guiding the choice of retreatment regimens [34].

### 7.3. Barriers to HCV Treatment Access in Europe

Treatment for HCV in Europe faces numerous barriers, which have limited the widespread use of highly effective DAAs.

As stated before, one of the most significant barriers has been the high prices of DAA therapies, leading to many restrictions set by payers and governments to manage the financial impact. Initially, DAAs were often only available to those with advanced liver disease (F3 and F4 stages), delaying treatment for patients with less severe disease [36,37].

Subsequently, governments have adopted a stepwise approach to DAA reimbursement, prioritizing patients with the most advanced liver disease and those at the greatest risk of complications. While this approach helped with managing costs, it restricted broader access to treatment. For example, in 2017, nearly half of the countries in the EU/EEA restricted DAAs to patients with F2 and above. Additionally, almost all countries required specialist prescription of DAAs, limiting access further. Efforts by France on the creation of a regional EU collective for price negotiations were not endorsed by other countries. So, the response to HCV has been largely country-specific, with considerable variability in how countries manage and fund HCV treatment. This variability led to significant differences in treatment coverage and access, with some countries like Liechtenstein and Switzerland prioritizing treatment for PWID regardless of substance use or liver fibrosis stage, while others maintain more restrictive policies [36,38,39].

Another issue has been the stigma associated with HCV, particularly among PWID, remaining a profound barrier. Many individuals with a history of drug use face discrimination within healthcare settings, which discourages them from seeking treatment. Despite the absence of restrictions in many European countries, some physicians hesitate to treat injecting drug users due to concerns about non-adherence, lower response rates, and the risk of reinfection [7,38].

Poor linkage to care is another critical barrier. The process from diagnosis to treatment often involves multiple outpatient visits, creating opportunities for patient drop-out. Administrative barriers and insufficient healthcare staffing exacerbate these challenges [36].

The COVID-19 pandemic has further compromised HCV treatment and diagnosis efforts. Lockdowns and healthcare system reallocations delayed diagnosis and treatment, impacting the progress to HCV elimination [36].

The persistence of risk of HCC after achieving SVR poses another challenge in patients with pre-existing advanced fibrosis or cirrhosis, with an annual incidence ranging from 2 to 2.5% across various studies [40], highlighting the need for maintained surveillance despite achieving SVR.

## 8. Public Health Strategies to Control HCV

There is not a unique pathway to HCV elimination in Europe and the strategies to meet the WHO targets should be tailored for each country, because every country has its own burden of HCV and its own organizational and financial contexts. In 2018, a review of the policies in 25 countries in Europe found that only 12 had a national strategy for HCV [41]. This approach is essential for patients with acute or chronic HCV infection who were not able to clear the infection and remain at risk of HCC and liver decompensation despite treatment with DAAs [36]. In most of the region, the goal should be increasing the number of people tested, diagnosed, and linked to care. Additionally, Razavi et al. showed that the rates of diagnosis should increase by two-fold in 28 European Countries to meet the WHO expectations [4].

Moreover, while HCV prevalence is expected to decrease, HCV-related morbidity and mortality are projected to rise in many countries unless significant improvements in diagnosis and treatment are made.

Despite this progress, fewer than half of European countries have a national HCV strategy. Each country must develop an individualized plan, guided by the WHO framework but tailored to their specific HCV burden and capabilities. The key to these plans will be the implementation of country-specific screening strategies to identify undiagnosed individuals and facilitate treatment [8,42].

Currently, the key components to reduce HCV prevalence include the needle and syringe programs and opioid substitution treatment as the primary prevention, along with the treatment of HCV with DAAs [12].

Efforts to eliminate HCV emphasize the importance of increasing treatment among PWID, because it will greatly reduce the transmission of the virus in most countries [36]. Prisons are also places where strategies to control HCV are very important due to the high prevalence in this setting. Also, epidemiological data of HCV in prisons in Europe are very scarce and efforts should be sought to improve them [36]. Spain has made significant progress through systematic screenings and universal treatment with DAAs. The JAILFREE-C program achieved an SVR in over 95% of treated inmates, leading to a significant reduction in HCV prevalence [43].

The reduction in DAA prices and increased discounts have facilitated broader treatment access. However, achieving WHO goals will require ongoing commitment and collaboration among governments, healthcare providers, and communities [8,42].

Also worth highlighting is the Hepcare Strategie. This is a project, supported by the EU and involving institutions in Ireland, United Kingdom, Spain, and Romania, focused on improving the care journey for individuals with HCV, particularly those who have experienced disruptions in care. The initiative tailors testing and treatment plans to local needs, incorporating peer support and community- and prison-based treatments, demonstrating that innovative, patient-centered approaches can effectively engage and retain patients in care. The project linked individuals to care, and a significant proportion began treatment and achieved SVR [39,44].

## 9. HCV Infection in Children in Europe

HCV infection in the pediatric population is also a significant public health concern with long-term implications in Europe. Globally, it is estimated that 3.26 million children aged 0–18 years are living with HCV, representing a prevalence of 0.13%. The burden of HCV among children varies significantly across regions, with Eastern Europe exhibiting the highest prevalence rates. The prevalence of HCV in women of childbearing age is the most significant predictor of HCV prevalence in children aged 0–4 years, and HCV prevalence in adults shows a strong correlation with HCV prevalence in children aged 5–19. Additionally, the proportion of HCV infections among PWIDs is associated with HCV prevalence in adolescents aged 15–19 years. As in other parts of the world, vertical transmission is the more frequent way of transmission [45].

Treatment options for pediatric HCV have historically been limited. The pre-DAA era primarily relied on interferon-based therapies, which were associated with low efficacy and high adverse effect profiles in children. The management and treatment practices for pediatric HCV in Europe were comprehensively surveyed in a study involving 38 pediatric specialists across 15 European countries in the pre-DAA era. The survey revealed that a significant proportion of children with HCV in Europe were untreated, largely due to the limitations of available therapies [46]. This scenario changed with the approval of DAAs for use in pediatric populations, offering a more effective and safer treatment option. Current ESPGHAN guidelines suggest that after 12 years old, all patients should be treated with a pan-genotypic regimen, and below 12 years, in most cases, treatment could be postponed until DAAs are approved for use in children between 3 and 11 years of age [47].

To achieve the World Health Organization’s HCV elimination targets by 2030, it is critical to also address the barriers to diagnosis and treatment in the pediatric population. This includes enhancing screening programs, particularly for women of childbearing age, and ensuring the availability and affordability of DAAs for children. Furthermore, there is a need for continued research to generate real-world evidence on the safety, efficacy, and acceptability of these treatments in pediatric patients.

## 10. Conclusions

HCV infection remains a significant public health challenge in Europe, with around 12 million people chronically infected. Additionally, despite the effectiveness of DAAs, high treatment costs, stigma, and poor linkage to care are barriers that compromise its efficacy. Geographic variations in prevalence, genotype distributions, and transmission routes further complicate management efforts.

Although the WHO aims to reduce liver-related deaths by 65%, new viral hepatitis infections by 90%, and ensure that 90% of infected individuals are diagnosed by 2030, most European countries face several barriers. Tailored public health strategies, including enhanced screening, financing, and strong political leadership, are crucial. Future directions involve ongoing research, new treatment developments, and addressing challenges, particularly among vulnerable populations.

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
