# Peer review of "Hepatitis C Virus Infection in Europe"

_pathogens, 2024, doi:10.3390/pathogens13100841_

Round 1

Reviewer 1 Report (Previous Reviewer 1)

Comments and Suggestions for Authors

I appreciate the efforts of the authors to improve the manuscript, which appears now more readable and focused to the point.

Only a minor point:

line 224: the risk of HCC is limited to patients with advanced fibrosis/cirrhosis. Please erase "particularly"

Author Response

Thank you very much for your comment. We agree with this observation. We have removed the word "particularly" from line 224 to clarify that the risk of HCC is limited to patients with advanced fibrosis/cirrhosis.

Reviewer 2 Report (New Reviewer)

Comments and Suggestions for Authors

First of all, I would like to thank the authors for the work they have done. This is a review study aimed at providing information on the status of HCV infection in Europe.
The authors provide epidemiological data from various European countries. As a suggestion for improvement, it might have been helpful to illustrate these figures through a table or figure that would allow for a quick comparison of prevalence rates across different countries.
They particularly reference the limited data on HCV in prisons. In this regard, I recommend reviewing the literature. There are quite a few articles on the reduction of hepatitis C in prisons. In Spain, over the last few years and following the introduction of direct-acting antivirals, there has been significant control and reduction of HCV prevalence through systematic screenings and universal treatments.
From a methodological point of view, they do not explain the steps followed in the development of this article. All review articles should include a methodology section where, even briefly, the processes of the bibliographic search, sources used, and criteria for inclusion and exclusion of selected articles are explained.
The following should be included within the methodology section:

  1. Type of review conducted.
  2. Bibliographic search: documentary sources; selected databases; search strategies; selection criteria for articles.
  3. Whether a critical reading of the articles selected for the review was carried out.
  4. Optional suggestion: add a table summarizing the articles selected for each subtopic discussed.

This would greatly enrich the article, giving it a more technical aspect and offering other authors insights on how to carry out an adequate review of the scientific literature.
The conclusions are consistent with the evidence and arguments presented, addressing the stated objectives. The authors have provided a European epidemiological overview of HCV, its transmission, diagnosis, current treatment guidelines, and main barriers.
The references are current and appropriate. It might be useful to include more references regarding the treatment in prisons, as there are several available on this topic.
Best regards.

Author Response

We would like to thank Reviewer 2 for the valuable feedback and thoughtful suggestions, which have helped to improve our manuscript. Below are our responses to each point raised:

Comment 1: As a suggestion for improvement, it might have been helpful to illustrate these figures through a table or figure that would allow for a quick comparison of prevalence rates across different countries.”

Response: We appreciate the suggestion to include a table for easier comparison of prevalence rates across different countries. However, since our review aims to provide a global overview of the HCV landscape, rather than focusing on specific countries or individual case studies, we have chosen not to add such a table. Our goal is to emphasize the broader epidemiological trends in HCV epidemiology and management across Europe.

Comment 2:”They particularly reference the limited data on HCV in prisons. In this regard, I recommend reviewing the literature. There are quite a few articles on the reduction of hepatitis C in prisons. In Spain, over the last few years and following the introduction of direct-acting antivirals, there has been significant control and reduction of HCV prevalence through systematic screenings and universal treatments.”

Response: We have revised the manuscript to include additional information about HCV in prison populations, specifically including data from Spain (line 265 to 268)

Comment 3: From a methodological point of view, they do not explain the steps followed in the development of this article. All review articles should include a methodology section where, even briefly, the processes of the bibliographic search, sources used, and criteria for inclusion and exclusion of selected articles are explained. The following should be included within the methodology section: Type of review conducted; Bibliographic search: documentary sources; selected databases; search strategies; selection criteria for articles; Whether a critical reading of the articles selected for the review was carried out.”

Response: We have added a methodology section (line 40 to 49) that details the type of review conducted (narrative review), the bibliographic search strategy (databases searched, keywords used, and inclusion/exclusion criteria), and the critical reading process of the selected literature. This addition strengthens the article’s technical depth and clarity.

This manuscript is a resubmission of an earlier submission. The following is a list of the peer review reports and author responses from that submission.

Round 1

Reviewer 1 Report

Comments and Suggestions for Authors

This is a narrative review on HCV in Europe, covering epidemiology, testing, therapy, and control strategies.

1.      The sections dealing with epidemiology are not clear. First of all, I suggest dividing data on incidence (newly or recently acquired infections) and prevalence. Incidence data, as defined above, are available for many countries and in different sub-populations. Note that prevalence studies have been performed mainly in selected populations (e.g., history of drug use, prisoners, pregnant women, hospital workers, etc.), which do not mirror the prevalence in the general population. When reporting these data, sample composition should be cited; extrapolating the data to the whole country/region should be avoided. There are very few studies sampling the general population; most of the data derive from estimates or modeling. I suggest reviewing the literature data and quoting original papers. For example, the data reported in reference 12 were obtained in a single area and do not represent the entire country. Again, reporting areas with 20% prevalence in Italy and Greece appears misleading.

2.      The figure of 15% awareness of their HCV infection (reference 4) seems obsolete. At line 78, it is reported that 38% of people with HCV in Europe have been diagnosed. At line 176, 26% of diagnosed individuals is reported. Please update and comment on these figures. Note also that salivary testing for anti-HCV is now widely used for screening.

3.      Therapy: Please note that telaprevir and boceprevir were administered with Peg-IFN, adding their own toxicity to that of IFN, resulting in high rates of adverse events. It should be considered that the eradication of HCV in patients with cirrhosis does not eliminate the risk of developing liver cancer (HCC). This explains why HCC will remain a health problem for a longer time.

Comments on the Quality of English Language

Acceptable

Author Response

Thank you very much for taking the time to review this manuscript and providing your thoughtful and constructive feedback. We have carefully considered each of the reviewers’ comments and have made the necessary revisions to improve the manuscript. Below, we provide a detailed response to each of the reviewers’ suggestions, highlighting the changes made to the manuscript.

REVIEWER 1:

Comment 1: The sections dealing with epidemiology are not clear. First of all, I suggest dividing data on incidence (newly or recently acquired infections) and prevalence. Incidence data, as defined above, are available for many countries and in different sub-populations. Note that prevalence studies have been performed mainly in selected populations (e.g., history of drug use, prisoners, pregnant women, hospital workers, etc.), which do not mirror the prevalence in the general population. When reporting these data, sample composition should be cited; extrapolating the data to the whole country/region should be avoided. There are very few studies sampling the general population; most of the data derive from estimates or modeling. I suggest reviewing the literature data and quoting original papers. For example, the data reported in reference 12 were obtained in a single area and do not represent the entire country. Again, reporting areas with 20% prevalence in Italy and Greece appears misleading.

Response:  We have revised the sections 2 (Epidemiology of HCV in Europe) and 3 (Geographical variations of HCV Infection in Europe) to clearly separate the data on incidence and prevalence (lines 40 to 48, 49 to 56,75 to 81 and 82 to 86). We have also reviewed and updated the literature, ensuring that the data are updated. The mention of prevalence in Italy and Greece has been reformulated to clearly indicate that there regions of the countries that have higher prevalences of HCV infection (lines 79 to 81).

Comment 2: The figure of 15% awareness of their HCV infection (reference 4) seems obsolete. At line 78, it is reported that 38% of people with HCV in Europe have been diagnosed. At line 176, 26% of diagnosed individuals is reported. Please update and comment on these figures. Note also that salivary testing for anti-HCV is now widely used for screening.

Response: The different prevalences through the manuscript reflected different studies and sources. We have updated the manuscript and the conflicting figures have been reconciled. The manuscript now reports the correct percentage based on the latest available data (line 25 to 29). Additionally, we have included the use of dried blood spot and saliva sampling as a screening method (line 132 to 134).

Comment 3: Please note that telaprevir and boceprevir were administered with Peg-IFN, adding their own toxicity to that of IFN, resulting in high rates of adverse events. It should be considered that the eradication of HCV in patients with cirrhosis does not eliminate the risk of developing liver cancer (HCC). This explains why HCC will remain a health problem for a longer time.

Response: We have revised the therapy section to clarify that the introduction of the first DAA and the first Interferon-free therapy occurred in different timings (line 151 to 155). We have also emphasized that HCV eradication does not eliminate the risk of hepatocellular carcinoma (HCC) in patients with cirrhosis and considerate it as one of the challenges that HCV elimination programs face (line 225 to 228).

Reviewer 2 Report

Comments and Suggestions for Authors

Your paper is interesting and the aim of your review in an european perspective should show the difference emerging in every country

Although the references are appropriate I think that for same issues could be useful to update your bibliography, for instance regarding the more recent extimation of people living with HCV infection

You can find more recent data and strategies on: 

Viral hepatitis B and C policies in countries and burden of disease in WHO regions, 2023 

Global hepatitis report 2024 (WHO); 

Securing wider EU commitment to the elimination of hepatitis C virus (Liver International. 2023;43:276–291)

regarding programme for eradication and national italian strategies :  Milestones to reach Hepatitis C Virus (HCV) elimination in Italy: From free-of-charge screening to regional roadmaps for an HCV-free nation (https://doi.org/10.1016/j.dld.2021.03.026)

and regarding the new strategies of italian public health 
R. D’Ambrosio et al., A territory-wide opportunistic, hospital-based HCV screening in the general population from Northern Italy. The 1969-1989 birth-cohort. EASL 2023

Author Response

Comment: Your paper is interesting and the aim of your review in an european perspective should show the difference emerging in every country Although the references are appropriate I think that for same issues could be useful to update your bibliography, for instance regarding the more recent extimation of people living with HCV infection

 You can find more recent data and strategies on:

Viral hepatitis B and C policies in countries and burden of disease in WHO regions, 2023 Global hepatitis report 2024 (WHO);

Securing wider EU commitment to the elimination of hepatitis C virus (Liver International. 2023;43:276–291)

regarding programme for eradication and national italian strategies :

Milestones to reach Hepatitis C Virus (HCV) elimination in Italy: From free-of-charge screening to regional roadmaps for an HCV-free nation (https://doi.org/10.1016/j.dld.2021.03.026)

and regarding the new strategies of italian public health

  1. D’Ambrosio et al., A territory-wide opportunistic, hospital-based HCV screening in the general population from Northern Italy. The 1969-1989 birth-cohort. EASL 2023

Response: The epidemiology section was reformulated to include recent references including the suggested ones (Reference 6 and 8).

Concernig the references about Italian strategies and programmes, we opted to not focus on a particular region or country. 

We believe these revisions have significantly improved the manuscript, and we hope it now meets the expectations of both the reviewers and the editorial team. We are grateful for the time and effort that the reviewers have invested in providing such valuable feedback.

Thank you for considering our revised manuscript for publication in Pathogens. We look forward to your favorable response.

Sincerely,